# Repurposing β-Lactams for the Treatment of *Mycobacterium kansasii* Infections: An In Vitro Study

**DOI:** 10.3390/antibiotics12020335

**Published:** 2023-02-05

**Authors:** Lara Muñoz-Muñoz, José A. Aínsa, Santiago Ramón-García

**Affiliations:** 1Department of Microbiology, Pediatrics, Radiology and Public Health, Faculty of Medicine, University of Zaragoza, 50009 Zaragoza, Spain; 2CIBER Respiratory Diseases (CIBERES), Health Institute Carlos III, 28029 Madrid, Spain; 3Research and Development Agency of Aragón (ARAID) Foundation, 50018 Zaragoza, Spain

**Keywords:** nontuberculous mycobacteria, *Mycobacterium kansasii*, repurposing, β-lactam combinations, amoxicillin–clavulanate, cefadroxil

## Abstract

*Mycobacterium kansasii* (*Mkn*) causes tuberculosis-like lung infection in both immunocompetent and immunocompromised patients. Current standard therapy against *Mkn* infection is lengthy and difficult to adhere to. Although β-lactams are the most important class of antibiotics, representing 65% of the global antibiotic market, they have been traditionally dismissed for the treatment of mycobacterial infections, as they were considered inactive against mycobacteria. A renewed interest in β-lactams as antimycobacterial agents has shown their activity against several mycobacterial species, including *M. tuberculosis*, *M. ulcerans* or *M. abscessus*; however, information against *Mkn* is lacking. In this study, we determined the in vitro activity of several β-lactams against *Mkn*. A selection of 32 agents including all β-lactam chemical classes (penicillins, cephalosporins, carbapenems and monobactams) with three β-lactamase inhibitors (clavulanate, tazobactam and avibactam) were evaluated against 22 *Mkn* strains by MIC assays. Penicillins plus clavulanate and first- and third-generation cephalosporins were the most active β-lactams against *Mkn*. Combinatorial time-kill assays revealed favorable interactions of amoxicillin–clavulanate and cefadroxil with first-line *Mkn* treatment. Amoxicillin–clavulanate and cefadroxil are oral medications that are readily available, and well tolerated with an excellent safety and pharmacokinetic profile that could constitute a promising alternative option for *Mkn* therapy.

## 1. Introduction

Infection by *Mycobacterium kansasii* (*Mkn*) is the second most prevalent cause of nontuberculous mycobacteria (NTM) disease in the United States, China, South America, and some European countries [1,2,3,4]. *Mkn* is one of the most virulent and prevalent NTM, being the most frequently found in immunocompetent patients [5]. In fact, because of its elevated pathogenicity, a single positive culture may be enough evidence to initiate treatment, while diagnosis of most NTM-caused disease needs isolation of bacteria from at least two temporal independent sputum samples [3,6].

*Mkn* produces a chronic fibrocavitary lung disease, mainly in the upper lobes, that resembles tuberculosis clinically and radiologically [1]. If untreated, it can cause extensive lung destruction and respiratory failure in 1 or 2 years [7]. Current treatment requires the combination of rifampicin with other antimicrobial drugs (i.e., ethambutol and isoniazid or ethambutol and clarithromycin) [3,4,6]. Guidelines recommend dosages of: rifampicin 600 mg/day, ethambutol 15 mg/kg/day, isoniazid 300 mg/day, and azithromycin–clarithromycin, 250 mg/day or 500 mg/day, respectively [3]. Therapy for *Mkn* pulmonary infection is challenging and it requires at least 12 months of a multidrug regimen to avoid the emergence of resistance and succeed in eradicating the infection [3,6]. These long combinatorial treatments often raise additional problems, including patient nonadherence and adverse events [3]. Thus, new alternatives are urgently needed to shorten the duration of *Mkn* therapies.

The β-lactams are one of the largest groups of antibiotics available today and have an excellent safety profile. Over the last 70 years, they have been used to treat most infections caused by Gram-positive or Gram-negative bacteria [8,9]. However, β-lactams have been traditionally dismissed to treat mycobacterial infections, due to the presence of constitutive β-lactamase enzymes and the low permeability of mycobacteria cell wall [10,11,12], although current knowledge might be challenging this paradigm [13]. Nevertheless, thanks to an increasing body of evidence from in vitro and in vivo studies and the clinic for tuberculosis and other mycobacterial infections, there is a renewed interest in β-lactams as antimycobacterial agents [10,11,14,15,16,17,18,19,20]. However, to date, there is no information on the potential role that β-lactams could play in *Mkn* therapy.

In this study, we report the in vitro activity of several β-lactams, including penicillins, cephalosporins, carbapenems and monobactams, against a panel of *Mkn* strains. From this initial screening, amoxicillin plus clavulanate and cefadroxil (two oral β-lactams) were selected for further evaluation in combination with first-line drugs for *Mkn* treatment, showing favorable in vitro interactions.

## 2. Results

### 2.1. Penicillins plus Clavulanate Together with First- and Third-Generation Cephalosporins Were the Most Active β-Lactams against Mkn

Thirty-two β-lactams, including penicillins, cephalosporins, carbapenems, a monobactam, and three β-lactamase inhibitors (clavulanate, tazobactam, and avibactam) were tested against the *Mkn* ATCC 12478 reference strain in both Middlebrook 7H9 and CAMHB base media. No major differences in β-lactam antimicrobial activity were observed between the two media used. Penicillins alone were inactive (MIC ≥ 64 mg/L); however, the addition of clavulanate substantially increased their activities, with MIC values of 8 mg/L. Some of the first- and third-generation cephalosporins showed MIC values of 8–32 mg/L, in contrast to second- and fourth-generation cephalosporins, which were inactive at values higher than 64 mg/L. The monobactam aztreonam and the carbapenems, except meropenem (MIC = 32 mg/L), were inactive (MIC ≥ 64 mg/L). The addition of tazobactam or avibactam had little or no effect on the antimicrobial activity of the β-lactam against *Mkn* (Table 1).

### 2.2. Amoxicillin–Clavulanate and Cefadroxil Were Validated as the Most Active β-Lactams against a Panel of Mkn Clinical Strains

As described above, clavulanate and avibactam were the most promising β-lactamase inhibitors. To further confirm this observation, 13 of the compounds representing the different subfamilies of β-lactams (based on in vitro data and clinical relevance) were tested in the presence of both clavulanate and avibactam against a panel of five *Mkn* clinical strains susceptible to current first-line *Mkn* antibiotics (Appendix A). While avibactam had little or no effect with most β-lactams, similar to previously observed, clavulanate significantly increased the activity of penicillins (penicillin, ampicillin, amoxicillin) with 4-fold MIC reductions against most strains. The activity of first (cefadroxil) and third(cefdinir) generation cephalosporins and meropenem was also observed against this panel of clinical strains; however, the addition of clavulanate or avibactam to these drugs had, in general, minor effects with ca. 2-fold reductions in MIC values. Aztreonam was ineffective in all conditions tested (MIC ≥ 64 mg/L) (Figure 1 and Appendix A).

Amoxicillin, cefadroxil and meropenem were thus selected for a third validation experiment with a panel of *Mkn* clinical strains (*n* = 21) (Appendix A). The combination amoxicillin–clavulanate was active in 19 out of the 21 clinical strains tested (MIC ≤ 32 mg/L) with a most frequent MIC value of 8–16 mg/L. Similarly, cefadroxil was active against all strains (MIC = 16–32 mg/L) and the presence of clavulanate further reduced MIC values to the 4–16 mg/L range. Clavulanate had no effect on the activity of meropenem, with a MIC of 32 mg/L (Figure 2 and Appendix A).

### 2.3. Time-Kill Assays Confirmed the Antimicrobial Activities of β-Lactams against Mkn

The activities of amoxicillin, cefadroxil and meropenem (alone and in the presence of clavulanate) were tested against the *Mkn* ATCC 12478 strain by dose-response time-kill assays (TKAs). As expected, amoxicillin alone had no inhibitory effect against *Mkn*, with only marginal bacteriostatic activity at the highest concentration tested (128 mg/L) and regrowth after only four days of incubation; however, addition of clavulanate markedly increased its antimycobacterial activity (i.e., similar bacteriostatic effects could be now observed at a low 0.5 mg/L concentration). In the presence of clavulanate, the highest concentration of amoxicillin prevented regrowth for up to 14 days, being the effect bacteriostatic (Figure 3A). Both cefadroxil and meropenem achieved a bactericidal activity over the time course (up to day 14 for both antibiotics and up to day 21 for cefadroxil) at the highest concentration tested (128 mg/L). The addition of clavulanate had, however, a marginal effect on the activity of meropenem, although it prevented regrowth observed at day 21. In the case of cefadroxil, the effect of clavulanate was limited, although a slight interaction could be observed at early time points and lower concentrations. To note that, under the concentration range tested, while meropenem displayed a clear cutoff concentration for antimicrobial activity (from 32 to 128 mg/L), in the case of cefadroxil the concentration range had a wider antibacterial effect, being bactericidal at high concentrations and bacteriostatic at concentrations from 8 to 32 mg/L (Figure 3B,C).

Time-kill kinetics were in agreement with previous MIC data. MIC values were based on growth inhibition with a growth limit of detection of ca. 10^7^ cells/mL [21] and determined after 6 days of incubation. For example, amoxicillin had an MIC value of ≥64 mg/L and of 8 mg/L when in combination with clavulanate (Table 1, Figure 1 and Figure 2), which correlates with the bacterial load in the time-kill assays at those concentrations at day 6 being below the limit of detection for MIC assays (Figure 3A). Similar observations were found for cefadroxil and meropenem (Figure 3B,C).

### 2.4. Role of Amoxicillin–Clavulanate and Cefadroxil in Combination with Current Standard Mkn Therapy

The contribution of amoxicillin–clavulanate and cefadroxil to the antimicrobial effect of current standard therapies for *Mkn* infections (i.e., rifampicin–ethambutol–isoniazid and rifampicin–ethambutol–clarithromycin) was evaluated by combinatorial time-kill assays against the *Mkn* ATCC 12478 strain (Figure 4, Figure 5 and Appendix A).

First, dose-response time-kill studies of the drugs alone were performed based on previous MIC values (Table 1). At 4xMIC, over the first seven days of incubation, amoxicillin–clavulanate (and to a lesser extend cefadroxil) displayed similar activity as clarithromycin, the most effective drug alone. After this period, a bacterial regrowth was observed due to β-lactam instability in the assay media (Figure 4A and Figure 5A). Then, both amoxicillin–clavulanate and cefadroxil were tested in pairwise combinations at matched xMIC values with each individual drug in the standard therapy. At 4x concentrations, amoxicillin–clavulanate strongly enhanced the activity of ethambutol (highly bactericidal over the first seven days), rifampicin (bacteriostatic activity compared to no activity of rifampicin alone) and clarithromycin (bacterial load down to the limit of detection at the latest time point of 21 days) (Figure 4B). Cefadroxil showed a strong synergism (clearly observed at 2x and 4x concentration) with all the compounds tested, the interaction with clarithromycin being the strongest (Figure 5B).

Finally, we evaluated how both β-lactams would integrate into standard treatment. While replacement of isoniazid or clarithromycin by either amoxicillin–clavulanate or cefadroxil was inferior to current recommended therapies (Appendix A), inclusion of amoxicillin–clavulanate or cefadroxil in a quadruple combination made it superior to the standard ones. The contribution of amoxicillin–clavulanate to the bactericidal activity of current therapies was better observed at 1x concentrations in the case of the isoniazid-containing regimen (faster bactericidal activity and extended regrowth prevention period) and at 2x concentrations in the case of the clarithromycin-containing regimen, where bacterial regrowth was prevented at the last 28-day time point (Figure 4C). The addition of cefadroxil implied a delay in regrowth in combination with rifampicin–ethambutol–isoniazid and similar activity as rifampicin–ethambutol–clarithromycin during the length of the experiment (Figure 5C).

## 3. Discussion

Despite being one of the most virulent and prevalent NTM, *Mkn* pulmonary infections are still cumbersome and treated with the classical multidrug regimen of rifampicin, ethambutol, and isoniazid for at least 12 months [3,6]. However, the role of isoniazid in *Mkn* treatment is currently under discussion, with some guidelines suggesting replacement of isoniazid with clarithromycin or azithromycin [4,6,22]. Our time-kill assays supported this recommendation: at 1x and 2x concentrations, the clarithromycin-containing combination was superior to the isoniazid-containing one, which was not able to prevent regrowth (Figure 4C and Figure 5C). Nevertheless, current *Mkn* therapy is lengthy and often associated with failure due to the lack of adherence to treatment or resistance to rifampicin or clarithromycin [3,23,24]. There is thus an urgent need to search for new agents to improve efficacy and decrease its duration [22]; however, current investment trends in *Mkn* treatment make it difficult for the development of a de novo specific *Mkn* antimicrobial [3]. As such, drug-repurposing strategies could bridge this development gap [25].

The β-lactams have historically been considered inactive against mycobacteria, due to three major factors: the poor cell permeability of the mycobacterial cell wall, the low affinity to penicillin-binding proteins, and especially the presence of constitutive β-lactamase enzymes [10]. First, the high lipid content of the cell wall reduces the permeability of β-lactam antibiotics to the inside of mycobacterial cells [26]. Second, β-lactams bind covalently to D,D-transpeptidases, belonging to the penicillin-binding protein (PBP) family, which are responsible for the intramolecular peptide 4:3 linkages between D-alanine–D-alanine in the peptidoglycan of most bacteria. Since the β-lactam ring is structurally similar to the D-alanine–D-alanine, D,D-transpeptidases may erroneously bind the β-lactam during cell wall synthesis. Thus, peptidoglycan synthesis stops, the cell wall weakens, permeability increases, and cell lysis occurs [27,28]. However, mycobacteria contain two types of peptidoglycan cross-linking: the well-known D,D-transpeptidases and L,D-transpeptidases that generate 3:3 linkages. L,D-transpeptidases are responsible for 80% formation of cross-links in mycobacterial peptidoglycan [8,12,29]. Finally, the major determinant of β-lactam resistance in mycobacteria is the constitutive presence of a β-lactamase enzyme that belongs to serine class A β-lactamases of the Ambler classification [10,11,12,30,31]. Indeed, it has been demonstrated that the addition of an inhibitor of this β-lactamase increases the activity of β-lactams in several mycobacterial species [10,32].

The implementation of two strategies has succeeded in reconsideration of β-lactams for mycobacterial therapy, due firstly to the development of new β-lactam antibiotics able to evade bacterial enzymatic inactivation conferred by β-lactamase, and secondly the discovery of β-lactamase inhibitors that block the β-lactamase and allow the associated β-lactam antibiotic to reach its target—the transpeptidases [28]. In recent years, thanks to these approaches, β-lactams have regained importance in the treatment of several mycobacterial species. Amoxicillin plus clavulanate and meropenem plus amoxicillin–clavulanate have been used against multidrug-resistant *M. tuberculosis* infections [10,33]. Moreover, amoxicillin–clavulanate is actually in clinical trials for the treatment of Buruli ulcer caused by *M. ulcerans* [34] and cefoxitin and imipenem are included in guideline recommendations in the treatment of *M. abscessus* [14,35]. However, there are hardly any data available on the activity of β-lactams against *Mkn*.

In this study, we explored the potential role of the β-lactams in *Mkn* therapy, reporting the most complete study of in vitro activity of β-lactams alone and in combination against *Mkn* to date. We performed an extensive screening of a panel of 32 β-lactams representing different β-lactam subfamilies against *Mkn* ATCC 12478 in two different media: Middlebrook 7H9 plus ADC and CAMHB plus OADC, which are recommended by EUCAST and CLSI, respectively [36,37]. No significant differences were found between the two media used, being amoxicillin–clavulanate (MIC = 8 mg/L) together with first- and third-generation cephalosporins (cefadroxil and cefdinir, MIC = 16–32 mg/L) and meropenem (MIC = 32 mg/L) the most active β-lactams (Table 1). The role of β-lactamase inhibitors in combination with some selected β-lactams was also studied against an extended panel of *Mkn* clinical isolates (Figure 1 and Figure 2, Appendix A). Clavulanate was the most effective, followed by avibactam, when combined with natural penicillins and aminopenicillins.

Clavulanate contains the classical β-lactam ring and tazobactam (a penicillin sulfone) possesses a β-lactam ring with minor modifications. On the contrary, avibactam contains a totally different structure, a diazabicyclooctane [8,28]. They also have a different mechanism of action: clavulanate and tazobactam bind irreversibly to the β-lactamase acting as a “suicide inhibitor”, whereas avibactam binds it reversibly [32]. Their activity spectrum against mycobacteria is also different. Clavulanate is active against slowly growing mycobacteria (SGM) such as *M. tuberculosis*, but it is inactivated by the BlaM β-lactamase of *M. abscessus*, a rapidly growing mycobacteria (RGM). By contrast, avibactam is active against RGM but not against the BlaC β-lactamase of *M. tuberculosis* [8]. RGM β-lactamases contain an SDN motif, whereas in SGM this motif is SDG, where glycine has replaced asparagine at the Ambler position 132. This substitution explains why RGM are more susceptible to avibactam (i.e., inhibition of BlaM in *M. abscessus*) while SGM are to clavulanic acid (i.e., irreversible inhibition of BlaC in *M. tuberculosis*) [8,11,38,39]. As *Mkn* belongs to SGM according to Runyon classification, our results are in agreement with these previous studies [5]. In our study tazobactam did not show activity: unlike clavulanate, sulfone inhibitors do not show activity against chromosomal β-lactamases [27].

Growth inhibition assays were secondary validated by time-kill assays (Figure 3). The antimycobacterial activity of amoxicillin (and cefadroxil to a lesser extent) was enhanced by the addition of clavulanate although the effect was not long-acting, possibly due to the rapid degradation of the β-lactams in the assay media. The half-lives of amoxicillin and clavulanate are approximately 7 and 2 days, respectively, while TKA are performed up to 28 days with no drug replacement [40]. Similarly, cefadroxil and meropenem concentrations are reduced by 90% and 62.9%, respectively, during the first 24 h of incubation [41,42].

Once the in vitro activity of the β-lactams against *Mkn* was confirmed by time-kill assays, we aimed at evaluating their potential role in combination with current anti-*Mkn* recommended treatments (Figure 4, Figure 5 and Appendix A). The standard therapy for treating *Mkn* infection is a multidrug regimen, to prevent the emergence of antibiotic resistance [3,6]. Adding an additional drug could be very advantageous in order to reduce the duration of the therapy and to make it more effective. A favorable interaction was observed between amoxicillin–clavulanate with either ethambutol, rifampicin and clarithromycin; cefadroxil, in addition, showed interaction with all the compounds tested, being the strongest with clarithromycin. These data are in agreement with previous reports in other mycobacterial species [9,10,29,43]. Pairwise interactions were dose-dependent and better observed at higher concentrations (2xMIC and 4xMIC). Importantly, while the inclusion of amoxicillin–clavulanate (or cefadroxil) to the rifampicin–ethambutol backbone was inferior compared to current triple therapies (Appendix A), addition to the current triple combinations (rifampicin–ethambutol–isoniazid and rifampicin–ethambutol–clarithromycin) showed a positive microbiological effect evidenced by lower CFU/mL counts and delayed regrowth profiles at the different concentrations tested (Figure 4 and Figure 5).

The β-lactams are one of the largest groups of antibiotics available today with an exceptional record of clinical safety in humans and no reported drug–drug interactions with drugs in standard therapy [44]. Used for decades to treat all sorts of bacterial infections, they were traditionally dismissed for the treatment of mycobacterial infections. Initial clinical studies with amoxicillin–clavulanate against mycobacteria gave discordant results [45,46]. Chambers et al. reported that the early bactericidal activity (EBA) of amoxicillin–clavulanate was comparable to that reported for antituberculous agents other than isoniazid when administering three daily doses [45]. However, Donald, PR et al. found that the EBA in patients receiving amoxicillin–clavulanic acid did not differ significantly from those receiving no drug when a single high dose was administered to the patients [46]. Differences in the time over the MIC (T > MIC) exposures might have explained these differences since T > MIC is the main pharmacokinetic driver of amoxicillin–clavulanate. Exposure is indeed an important factor in β-lactam therapy, including drug development since it compromises in vivo studies in mice [19]. In a seminal clinical study for TB treatment, oral faropenem showed no significant EBA activity, while the intravenous meropenem in combination with amoxicillin–clavulanate was comparable to the standard therapy [47]. However, although effective in some controlled context, side effects might be a concern for the use of intravenous meropenem [16,48]. Due to the lengthy *Mkn* treatment (up to 12 months), oral alternatives need to be found to improve patient compliance.

Amoxicillin–clavulanate continues to be one of the most widely used antibiotics for clinical use and it is commercially available in several oral formulations [49]. It is commonly used in the treatment of respiratory infections. Penetration of amoxicillin into the respiratory tract is greater than, for example, ampicillin despite similar peak serum concentrations. The usual single oral doses of 500/125 mg and 875/125 mg of amoxicillin–clavulanate (C_max_ values of 7.2 mg/L and 11.6 mg/L, respectively) would achieve the antimycobacterial concentration reported in our studies (MIC = 8 mg/L), although higher dosed might be needed to reach clinically effective exposures. For example, for the treatment of *S. pneumoniae* strains with MIC values of 4 mg/L, amoxicillin–clavulanate doses of 875/125 mg and 1000/125 mg three times daily were effective; however, with MIC values of 8 mg/L higher doses of 2000/125 mg twice daily were needed for activity [49].

As shown in our study (Figure 4 and Figure 5), these pharmacological limitations might be compensated when used in synergistic combinations, similar to current clinical trials for the treating of *M. ulcerans* [34]. In fact, several β-lactams displayed synergistic interactions with rifampicin against *M. tuberculosis* and *M. ulcerans* [9,43], with ethambutol against *M. fortuitum*, *M. marinum*, *M. tuberculosis* and *Mkn* [9,10], with new antituberculosis drugs such as bedaquiline, delamanid and pretomanid against *M. tuberculosis*, and with clarithromycin against *M. ulcerans* [9].

Cefadroxil, cephalexin, cephradine, cefdinir and cefditoren are also administered orally, although cefdinir and cefditoren have low absorption levels [43,50,51]. In our studies, cefadroxil, cephalexin and cephradine showed similar in vitro activities against *Mkn* and pharmacological exposure could be higher with cefadroxil [52]. Cefadroxil could be a suitable option as well: a single oral dose of 500 mg would reach plasma concentrations of 16 mg/L, while a dose of 300 mg of cefdinir would only achieve 2.84 mg/L and 400 mg of cefditoren 4.6 mg/L [9,11,51].

This is the first study comprehensively reporting the activity of β-lactams alone and in combination against *Mkn*; however, it has some limitations. Our study is limited to extracellular, planktonic growth conditions. Other models reflecting the physiology of the bacteria at the site of infection such as slow or nonreplicating, intracellular or biofilm growth conditions would contribute to a more complete data set [53]. In addition, the concentrations used in the time-kill assays were selected based on microbiological endpoints (MIC values) to evaluate the degree of in vitro interactions. Future studies will need to address the pharmacokinetic/pharmacodynamic (PK/PD) properties or serum concentrations reached at the site of infection of the compounds [21]. Novel modeling strategies coupled with dynamic in vitro PK/PD models, such as the hollow fiber system, might help to inform future clinical trials [54,55,56].

In conclusion, our data together with mounting evidence on the potential role of β-lactams as antimycobacterial agents should promote further research in this area to explore optimal β-lactam-containing combinations to improve current anti-*Mkn* therapeutic options. The β-lactams have an excellent safety record and are available in numerous formulations, being an alternative option for potential inclusion in *Mkn* therapy [47]. Amoxicillin–clavulanate and cefadroxil would be the most promising against *Mkn*, with the best pharmacological properties including oral bioavailability. Moreover, amoxicillin–clavulanate and cefadroxil are well tolerated, and there are no described pharmacological drug–drug interactions with current recommended standard treatments [49,52].

## 4. Materials and Methods

### 4.1. Mycobacterial Strains

Twenty-two *Mkn* strains were used in this study. The ATCC 12478 reference strain was procured from the American Type Culture Collection. Additionally, 21 clinical isolates were provided by the Microbiology Department of Lozano Blesa Clinical University Hospital, Zaragoza, Spain. *Mkn* clinical isolates were identified using GenoType *Mycobacterium* CM/AS assay (Hain Lifescience GmbH, Nehren, Germany) and PCR-restriction fragment length polymorphism analysis [57].

### 4.2. Culture Media

Mycobacterial strains were tested in two different media. Middlebrook 7H9 broth (Difco, Beirut, Lebanon) supplemented with 10% albumin, dextrose, and catalase (ADC) (Difco), and 0.5% glycerol (Scharlab, Barcelona, Spain), as recommended by EUCAST [36], and Cation Adjusted Mueller–Hinton Broth (CAMHB) supplemented with 5% oleic acid, albumin, dextrose, and catalase (OADC) (Difco), as recommended by CLSI [37]. Middlebrook 7H10 agar plates (Difco) supplemented with 10% OADC (Difco), 0.2% glycerol, 0.4% activated charcoal (Sigma Aldrich, St. Louis, MO, USA) and 0.05% tween (Scharlab) were used for bacterial CFU enumeration.

### 4.3. Antibiotics

Thirty-two β-lactams were evaluated, including penicillins, cephalosporins, carbapenems and a monobactam. Penicillins comprised penicillin G (Sigma), ampicillin (Sigma), amoxicillin (Sigma), piperacillin (European Pharmacopeia), cloxacillin (European Pharmacopeia) and oxacillin (European Pharmacopeia). Cephalosporins comprised cefadroxil (Sigma), cephalexin (Sigma), cefazolin (Medicinal Chemistry), cephradine (Sigma), cefoxitin (European Pharmacopeia), cefonicid (Sigma), cefamandole (Medicinal Chemistry), cefotiam (Sigma), cefuroxime (European Pharmacopeia), cefotaxime (European Pharmacopeia), ceftriaxone (European Pharmacopeia), cefdinir (Sigma), cefditoren (Medicinal Chemistry), cefcapene (Sigma), cefixime (Medicinal Chemistry), cefpodoxime (Medicinal Chemistry), ceftiofur (Sigma), ceftazidime (European Pharmacopeia), cefpirome (Sigma) and cefepime (European Pharmacopeia). Carbapenems comprised imipenem (Sigma), meropenem (Kabi), ertapenem (MSD), doripenem (Sigma) and faropenem (Sigma). The monobactam was aztreonam (Sigma). β-lactamase inhibitors were also included in the study: clavulanate (Sigma), tazobactam (Sigma) and avibactam (Adooq). As well as drugs used in standard treatment: rifampin (Sigma), isoniazid (Fluka), ethambutol (Sigma) and clarithromycin (Sigma).

Antibiotics were dissolved according to the manufacturer’s instructions. β-lactams and β-lactamase inhibitors were always prepared fresh on the same day of plate inoculation. Standard treatment antibiotics were prepared in a stock solution (10 mg/mL), aliquoted and stored at −20 °C until further use.

### 4.4. Antibiotics Susceptibility Testing

MIC determinations were performed by broth microdilution assays in a 96-well plate [58]. Antibiotics were manually transferred to 96-well plates and serially twofold diluted. Then, mycobacterial cells were inoculated to each well to achieve a final density of 5 × 10^5^ cells/mL. Positive and negative growth controls were included in every plate. Plates were incubated at 37 °C for five days. Then, 30 µL of redox indicator 3-(4,5-dimethylthiazol-2-yl)-2,5-diphenyl tetrazolium bromide (MTT) was added to the wells and further incubated overnight. Optical density (OD) was read at 580 nm to measure the MTT to formazan conversion.

For calculating the MIC, we transformed the OD values into percentage of MTT conversion for each well using the equation below (1). The MIC was defined as the lowest concentration at which percentage of growth was ≤10%. Experiments were performed in triplicate and repeated at least twice.
%MTT conversion = [(OD_1_ ×100)/(OD_2_)] − [(OD_3_ × 100)/(OD_2_)](1)

OD_1:_ optical density at 580 nm of the test well (with antibiotic treatment). OD_2:_ average optical density at 580 nm of the untreated control well (100% growth). OD_3:_ average optical density at 580 nm of the uninoculated, untreated controls (background signal, 0% growth).

### 4.5. Time-Kill Assays

Mycobacterial cultures were prepared to a final cell density of 10^5^ cells/mL and 2.5 mL of this pre-inoculum was transferred to each well of a 24-well plate. Different concentrations of the antibiotics were added to each well, as appropriate. Plates were incubated at 37 °C for 21 or 28 days. At every time point, 20 µL of each condition was removed from the 24-well plates and serially 10-fold diluted in 180 µL of phosphate-buffered saline (PBS) (Millipore) with 0.1% tyloxapol (Sigma). Then, 2.5 µL of each dilution (including undiluted samples) were plated on 7H10 agar plates. CFU were determined by colony counting after 12 days of incubation. Plates were checked again one and two weeks later for late growers. Experiments were performed in duplicate and repeated at least twice.

## Figures and Tables

**Figure 1 antibiotics-12-00335-f001:**
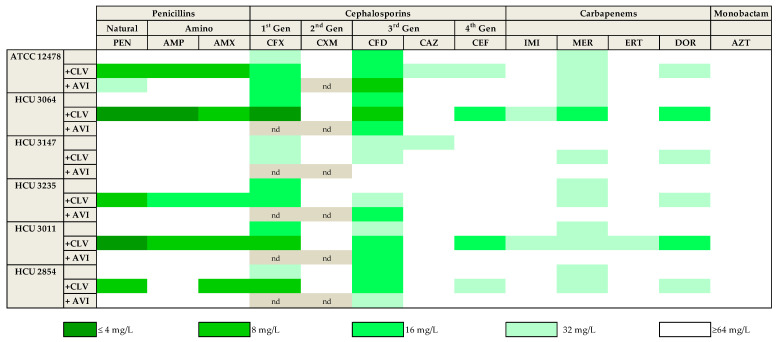
Heat map representation of the activity of thirteen selected β-lactams against *Mkn* clinical strains. MIC values (mg/L) were calculated in the presence/absence of a fixed 4 mg/L dose of clavulanate and avibactam in Middlebrook 7H9 broth plus ADC. Clavulanate was the most effective β-lactamase inhibitor. Amoxicillin–clavulanate and cefadroxil were the more active beta-lactams. PEN: penicillin; AMP: ampicillin; AMX: amoxicillin. CFX: cefadroxil; CXM: cefuroxime; CFD: cefdinir; CAZ: ceftazidime; CEF: cefepime. IMI: imipenem; MER: meropenem; ERT: ertapenem; DOR: doripenem. AZT: aztreonam. CLV: clavulanate; AVI: avibactam. nd: not determined.

**Figure 2 antibiotics-12-00335-f002:**
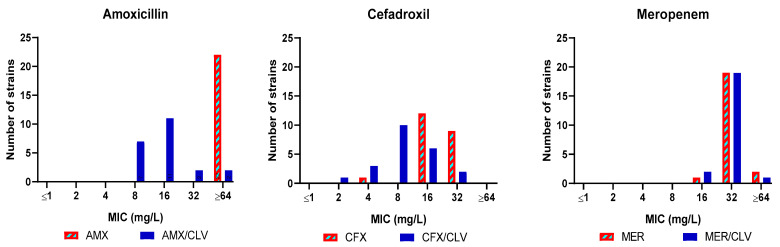
MIC distribution of amoxicillin, cefadroxil and meropenem tested in presence/absence of clavulanate against a panel of *Mkn* clinical isolates. MIC (mg/L) values were determined in Middlebrook 7H9 broth plus ADC. Clavulanate was added at a fixed 4 mg/L dose. AMX: amoxicillin; CFX: cefadroxil; MER: meropenem; CLV: clavulanate.

**Figure 3 antibiotics-12-00335-f003:**
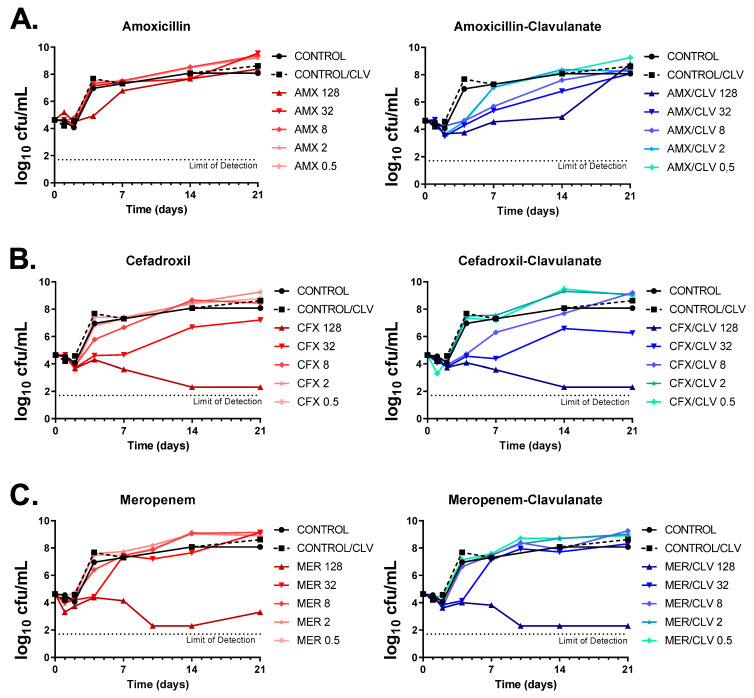
Dose-response time-kill assays of β-lactams against *Mkn*. (**A**) Amoxicillin, (**B**) cefadroxil and (**C**) meropenem were tested alone and in combination with clavulanate against the *Mkn* ATCC 12478 strain in Middlebrook 7H9 broth plus ADC. Two positive growth control cultures were included: one with no antibiotics (CONTROL) and one containing only clavulanate (CONTROL/CLV). Data represent one experiment of at least two independent experiments performed in duplicate. Clavulanate was added at a fixed dose of 4 mg/L. AMX: amoxicillin; CFX: cefadroxil; MER: meropenem; CLV: clavulanate.

**Figure 4 antibiotics-12-00335-f004:**
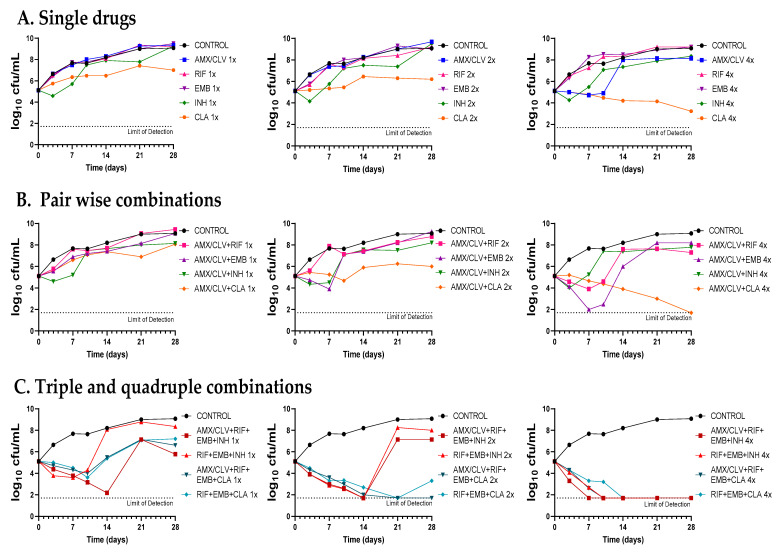
Time-kill assays of amoxicillin–clavulanate in combination with standard therapies against *Mkn* ATCC 12478. (**A**) Single drugs. (**B**) Pairwise combinations of amoxicillin–clavulanate with either rifampicin, ethambutol, isoniazid or clarithromycin. (**C**) Standard triple therapies against *Mkn* and quadruple combinations including amoxicillin–clavulanate. Time-kill assays were performed in Middlebrook 7H9 broth plus ADC. MIC values used were: AMX: 8 mg/L; RIF: 0.125 mg/L; EMB: 4 mg/L; CLA: 0.25 mg/L and INH: 8 mg/L. Clavulanate was added at a fixed dose of 4 mg/L. A drug-free positive control culture was included (CONTROL). Data represents one experiment of at least two independent experiments performed in duplicate. AMX/CLV: amoxicillin–clavulanate; RIF: rifampicin, EMB: ethambutol, INH: isoniazid, CLA: clarithromycin.

**Figure 5 antibiotics-12-00335-f005:**
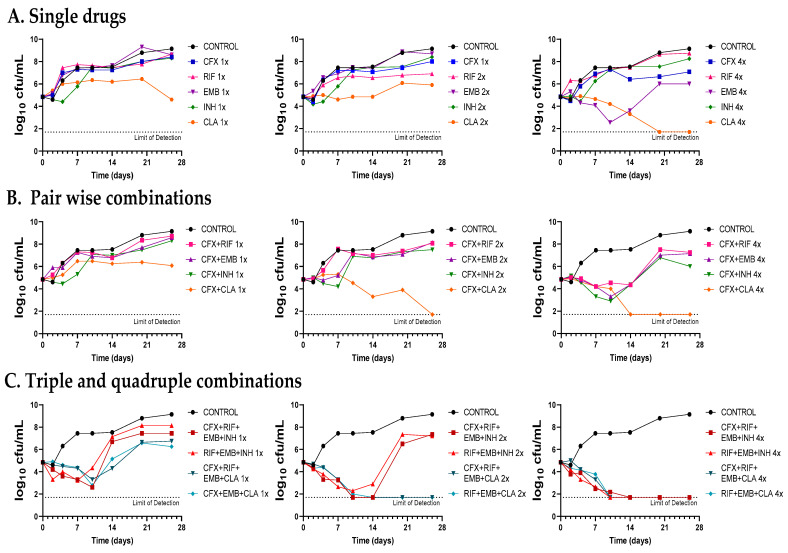
Time-kill assays of cefadroxil in combination with standard therapies against *Mkn* ATCC 12478. (**A**) Single drugs. (**B**) Pairwise combinations of cefadroxil with either rifampicin, ethambutol, isoniazid or clarithromycin. (**C**) Standard triple therapies against *Mkn* and quadruple combinations including cefadroxil. Time-kill assays were performed in Middlebrook 7H9 broth plus ADC. MIC values used were: CFX: 8 mg/L; RIF: 0.125 mg/L; EMB: 4 mg/L; CLA: 0.25 mg/L and INH: 8 mg/L. A positive control with no antibiotics was included (CONTROL). Data represent one experiment of at least two independent experiments performed in duplicate. CFX: cefadroxil; RIF: rifampicin, EMB: ethambutol, INH: isoniazid, CLA: clarithromycin.

**Table 1 antibiotics-12-00335-t001:** MIC (mg/L) of β-lactams against the *Mkn* ATCC 12478 strain. Penicillins, cephalosporins, carbapenems and a monobactam in the presence or absence of β-lactamase inhibitors were assayed against the *Mkn* ATCC 12478 reference strain in Middlebrook 7H9 broth plus ADC and cation adjusted Mueller–Hinton broth plus OADC. ^1^ β-lactamase inhibitors were tested at a fixed concentration of 4 mg/L.

		MIC (mg/L) against *Mkn* ATCC 12478		
Chemical Class		Compound	7H9/ADC	CAMHB/OADC
Penicillins	Natural penicillin	Penicillin	≥64	≥64
		Penicillin–clavulanate ^1^	8	nd
		Penicillin–tazobactam ^1^	≥64	nd
		Penicillin–avibactam ^1^	32	nd
	Aminopenicillin	Ampicillin	≥64	≥64
		Ampicillin–clavulanate ^1^	8	nd
		Ampicillin–tazobactam ^1^	≥64	nd
		Ampicillin–avibactam ^1^	32	nd
		Amoxicillin	≥64	≥64
		Amoxicillin–clavulanate ^1^	8	8–16
		Amoxicillin–tazobactam ^1^	≥64	nd
		Amoxicillin–avibactam ^1^	32	nd
	Ureidopenicillin	Piperacillin	≥64	≥64
		Piperacillin–tazobactam ^1^	≥64	≥64
	Penicillinase-resistant penicillin	Cloxacillin	≥64	≥64
		Oxacillin	≥64	≥64
Cephalosporins	1st-Generation cephalosporins	Cefadroxil	16–32	32–64
		Cefadroxil–clavulanate ^1^	16	nd
		Cefadroxil–tazobactam ^1^	16	nd
		Cefadroxil–avibactam ^1^	16	nd
		Cephalexin	32	≥64
		Cefazolin	≥64	≥64
		Cephradine	8–16	≥64
	2nd-Generation cephalosporins	Cefoxitin	≥64	≥64
		Cefocinid	≥64	≥64
		Cefamandole	≥64	≥64
		Cefotiam	≥64	≥64
		Cefuroxime	≥64	≥64
	3rd-Generation cephalosporins	Cefotaxime	≥64	≥64
		Ceftriaxone	≥64	≥64
		Cefdinir	16	32–64
		Cefdinir–clavulanate ^1^	16	nd
		Cefdinir–tazobactam ^1^	16	nd
		Cefdinir–avibactam ^1^	8	nd
		Cefditoren	16	32
		Cefcapene	≥64	≥64
		Cefixime	≥64	≥64
		Cefpodoxime	32	≥64
		Ceftiofur	16–32	16
		Ceftazidime	≥64	≥64
	4th-Generation cephalosporins	Cefpirome	≥64	≥64
		Cefepime	≥64	≥64
		Cefepime–clavulanate ^1^	32	nd
		Cefepime–tazobactam ^1^	nd	nd
		Cefepime–avibactam ^1^	≥64	nd
Carbapenems		Imipenem	≥64	≥64
		Imipenem–clavulanate ^1^	≥64	nd
		Imipenem–tazobactam ^1^	≥64	nd
		Imipenem–tazobactam ^1^	≥64	nd
		Meropenem	32	32–64
		Meropenem–clavulanate ^1^	32	nd
		Meropenem–tazobactam ^1^	nd	nd
		Meropenem–tazobactam ^1^	32	nd
		Ertapenem	≥64	≥64
		Ertapenem–clavulanate ^1^	≥64	nd
		Ertapenem–tazobactam ^1^	nd	nd
		Ertapenem–tazobactam ^1^	≥64	nd
		Doripenem	≥64	≥64
		Doripenem–clavulanate ^1^	32	nd
		Doripenem–tazobactam ^1^	≥64	nd
		Doripenem–tazobactam ^1^	≥64	nd
		Faropenem	≥64	32–64
Monobactam		Aztreonam	≥64	≥64

## Data Availability

Not applicable.

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
