# Peer review of "Repurposing β-Lactams for the Treatment of Mycobacterium kansasii Infections: An In Vitro Study"

_antibiotics, 2023, doi:10.3390/antibiotics12020335_

Round 1

Reviewer 1 Report

The manuscript submitted by Muñoz-Muñoz et al reports the in vitro activity of 32 b-lactams as a potential treatment for Mycobacterium kanasii. This manuscript is well structured overall, but it needs minor modifications, that I have outlined in the comments below.

1.     Introduction section: Please elaborate more about Mycobacterium kanasii.

2.     Please mention the dosage of rifampin, isoniazid, and ethambutol in the current approved treatment.

3.     Line 65: Why penicillin was found inactive in the assay, whereas combination of clavulanate showed good activity? Did the authors try clavulanate alone?

4.     Line 80: Readers would be interested to know the mechanism of clavulanate and avibactam being the most promising b-lactamase inhibitors.

5.     Figure 1: The authors have to explain in detail what figure is all about and take-home message.

6.     Figure 2: Why rifampin/ or combinations was not used as a control? Please explain

7.     Line 131: Figure 3A? Can’t see 3A

8.     Figure 3: Please label as Figure 3A, 3B and 3C also what was the control? Please mention.

9.     Figure 4: What is the rationale behind choosing quadruple combination? Again, what control was used?

10.  Line 299 and 307: Use only conclusion (remove summary word from the line 307)

11.  What if b-lactams has the same side effects as of rifampin? Do authors have any thoughts?

12.  The authors have to proofread the manuscript, there are grammatical errors

Author Response

Reviewer 1

Point 1.1: Introduction section: Please elaborate more about Mycobacterium kansasii.

Answer 1: We have added information about Mkn and the type of infection that cause and consequences. 

“Mkn produces a chronic fibrocavitary lung disease, mainly in the upper lobes, that resembles clinically and radiologically to tuberculosis [1]. If untreated it can cause extensive lung destruction and respiratory failure in 1 or 2 years [7]. Current treatment requires the combination of rifampicin with other antimicrobial drugs (i.e., ethambutol and isoniazid or ethambutol and clarithromycin) [3,4,6]. Guidelines recommend dosages are: rifampicin, 600mg/day; ethambutol, 15mg/kg/day; isoniazid, 300mg/day and; azithromycin/clarithromycin, 250mgmg/day or 500mg/day, respectively [3] Therapy for Mkn pulmonary infection is challenging and it requires at least 12 months of a multi-drug regimen to avoid the emergence of resistance and succeed in eradicating the infection [8,9]. These long combinatorial treatments often raise additional problems, including patient non-adherence and adverse events [3]. Thus, new alternatives are urgently needed to shorten the duration of Mkn therapies.” (Lines 40-51)

Point 1.2: Please mention the dosage of rifampin, isoniazid, and ethambutol in the current approved treatment. 

Answer 2: We have included dosage data.

See point 1”.

Point 1.3: Line 65: Why penicillin was found inactive in the assay, whereas combination of clavulanate showed good activity? Did the authors try clavulanate alone? 

Answer 3: The reason why clavulanate enhances penicillin activity is now included in the discussion. Basically, it prevents penicillin from being destroyed by the natural mycobacterial defenses (Lines 231-253). In addition, clavulanate alone was tested in time-kinetics assays (CONTROL/CLV) and as expected it develops the same behaviour as control with no antibiotics (CONTROL). We have described it in more detail in the Figure 3 caption (Lines 154-160).

Point 1.4: Line 80: Readers would be interested to know the mechanism of clavulanate and avibactam being the most promising b-lactamase inhibitors. 

Answer 4: The mechanism is now explained in the discussion (Lines 274-289)

Point 1.5: Figure 1: The authors have to explain in detail what figure is all about and take-home message. 

Answer 5: The legend of the figure has now been modified. 

Figure 1. Heat map representation of the activity of thirteen selected β-lactams against Mkn clinical strains. MIC values (mg/L) were calculated in the presence/absence of a fixed 4 mg/L dose of clavulanate and avibactam in Middlebrook 7H9 broth plus ADC. Clavulanate was the most effective β-lactamase inhibitor. Amoxicillin/clavulanate and cefadroxil were the more active beta-lactams. PEN: penicillin; AMP: ampicillin; AMX: amoxicillin. CFX: cefadroxil; CXM: cefuroxime; CFD: cefdinir; CAZ: ceftazidime; CEF: cefepime. IMI: imipenem; MER: meropenem; ERT: ertapenem; DOR: doripenem. AZT: aztreonam. CLV: clavulanate; AVI: avibactam. nd: not determined” (Lines 104-110)

Point 1.6: Figure 2: Why rifampin/ or combinations was not used as a control? Please explain 

Answer 6: All strains included in the manuscript (also those of figure 2) were tested also against control antibiotics of the standard therapy (rifampicin, ethambutol, isoniazid and clarithromycin). 

We have included a supplementary Table S2 showing this MIC data. 

Amoxicillin, cefadroxil and meropenem were thus selected for a third validation experiment with a panel of Mkn clinical strains (n= 21) (Table S2). The combination of amoxicillin/clavulanate was active in 19 out of the 21 clinical strains tested (MIC ≤32 mg/L) with a most frequent MIC value of 8-16 mg/L. Similarly, cefadroxil was active against all strains (MIC= 16-32 mg/L) and the presence of clavulanate further reduced MIC values to the 4-16 mg/L range. Clavulanate had no effect on the activity of meropenem, with a MIC of 32 mg/L (Figure 2 and Table S2).” (Lines 112-118)

Legend of Figure 2 has also been updated: 

“Figure 2. MIC distribution of amoxicillin, cefadroxil and meropenem test-ed in presence/absence of clavulanate against a panel of Mkn clinical isolates. MIC (mg/L) values were determined in Middlebrook 7H9 broth plus ADC. Clavulanate was added at a fixed 4 mg/L dose. AMX: amoxicillin; CFX: cefadroxil; MER: meropenem; CLV: clavulanate.” (Lines 121-124)

Point 1.7: Line 131: Figure 3A? Can’t see 3A 

Answer 7: We have split the Figure 3 as 3A, 3B and 3C and referenced each figure as appropriate in the text (Lines 128-152)

Point 1.8: Figure 3: Please label as Figure 3A, 3B and 3C also what was the control? Please mention. 

Answer 8: We have labelled the Figure as 3A, 3B and 3C, also the explanation of the control has been added.

Figure 3. Dose-response time-kill assays of β-lactams against Mkn. (A) Amoxicillin, (B) cefadroxil and (C) meropenem were tested alone and in combination with clavulanate against the Mkn ATCC 12478 strain in Middlebrook 7H9 broth plus ADC. Two positive growth control cultures were included: one with no antibiotics (CONTROL) and one containing only clavulanate (CONTROL/CLV). Data represent one experiment out of, at least, two independent experiments performed in duplicates. Clavulanate was added at a fixed dose of 4 mg/L. AMX: amoxicillin; CFX: cefadroxil; MER: meropenem; CLV: clavulanate. (Lines 154-160)

Point 1.9: Figure 4: What is the rationale behind choosing quadruple combination? Again, what control was used? 

Answer 9:  It is now included in the discussion. 

The standard therapy for treating Mkn infection is already a multidrug regimen, to prevent the emergence of antibiotic resistance [3,6]. Therefore, adding an additional drug would not make a big difference to the current treatment, and it could be very advantageous if we achieve to reduce the duration of the therapy and make it more effective.” (Lines 300-303)

Point 1.10: Line 299 and 307: Use only conclusion (remove summary word from the line 307) 

Answer 10: We have combined the two paragraphs together. It now reads:

In conclusion, our data together with mounting evidence on the potential role of β-lactams as anti-mycobacterial agents should promote further research in this area to explore optimal β-lactam-containing combinations to improve current anti-Mkn therapeutic options. β-lactams have an excellent safety record and they are available in numerous formulations, being an alternative option for potential inclusion in Mkn therapy [57]. Amoxicillin/clavulanate and cefadroxil would be the most promising against Mkn, with the best pharmacological properties including oral bioavailability. Moreover, amoxicillin/clavulanate and cefadroxil are well tolerated, and there are no described pharmacological drug-drug interactions with current recommended standard treatments [40,47].” (Lines 371-379)

Point 1.11: What if b-lactams has the same side effects as of rifampin? Do authors have any thoughts? 

Answer 11: β-lactams are extraordinarily well tolerated. The most common adverse effects are gastrointestinal events and hypersensitivity reasons. Rifampicin, isoniazid and clarithromycin also have gastrointestinal effects, but they also present more serious problems. Rifampicin induces hepatitis, anemia and the ‘‘flu-like syndrome’’ (fever, thrombocytopenia, and renal failure). Isoniazid can develop peripheral neuropathy, drug-induced lupus erythematosus and hepatotoxicity. Moreover, rifampicin and isoniazid interfere with the cytochrome P450 that can result in a decrease/increase of serum concentrations of the concurrent drugs. Clarithromycin can prolong the QT interval in the cardiac cycle, with risk of cardiac arrhythmias. Moreover, clarithromycin can cause sensorineural hearing loss, Stevens-Johnsons syndrome and toxic epidermal necrolysis. Ethambutol cause optic neuritis, special care must be taken when administering it to children. All things consider, β-lactams have fewer adverse effects than the antibiotics including in the current treatment. In fact, standard Mkn therapy in mild pulmonary diseases is recommended to be given intermittently 3-days weekly rather than daily to avoid toxicity.

We have added a sentence:

“The β-lactams are one of the largest groups of antibiotics available today with an exceptional record of clinical safety in humans and no reported drug-drug interactions with the drugs in the standard therapy [47]” (Lines 315-317)

Point 1.12:  The authors have to proofread the manuscript, there are grammatical errors

Answer 12: Grammatical errors were corrected

Reviewer 2 Report

Dear respected editor;

Regarding the article number Antibiotics-2162038, titled; Repurposing β-lactams for the treatment of Mycobacterium kansasii infections: an in vitro study

This study aimed to evaluate the in vitro activity of several β-lactams, including penicillins, cephalosporins, carbapenems and monobactams, against a panel of Mkn strains. In addition, two oral  β-lactams were further evaluated in combination with first-line drugs for Mkn treatment, showing in vitro favorable interactions

General points to be considered

-          Title: good  

-          Abstract: should be corrected

-          Authors should minimize the introduction and illustrates more about results and also, they have to rewrite the conclusion.

Line 13-15 in the second sentence should be corrected (rewrite it). β-lactams antibiotics are the most important class of …………………………………………………………………………β-lactams are the most important class of antibiotics with an estimated of 65% of the global antibiotic market; although, they have been traditionally dismissed for the treatment of mycobacterial infections, as they were considered inactive against mycobacteria.

-          All the citation formats of the references in the text should be corrected according to the journal guidelines.

1.     Introduction:

The first sentence lines 30-34 should be split into two sentences and rewritten. Also, the format style of references should be corrected according to journal style [1-4].

Mycobacterium kansasii (Mkn) is one of the most virulent and prevalent non-tuberculous mycobacteria (NTM), the most frequently found in immunocompetent patients and, in general, the second most prevalent cause of NTM disease in the United States, China, South American and some European countries [1]–[4].

-          The second sentence lines 33-35, also needs correction and rephrasing.

-          While diagnosis of most NTM- caused disease needs isolation of bacteria from at least two temporal independent sputum samples, because of the elevated pathogenicity of Mkn, a single positive culture may be enough evidence to initiate treatment.

-          Lines 44-46, should be corrected grammatically and the references should be corrected also.

-          β-lactams are one of the largest groups of antibiotics available today with an excellent safety profile; they have been used to treat most infections caused by Gram-positive and 4 Gram-negative bacteria for the last 70 years [6], [7].

-          The objectives of the study need to be rewritten and connecting words should be include for the second objective.

-          In this study, we report the in vitro activity of several β-lactams, including penicillins, cephalosporins, carbapenems and monobactams, against a panel of Mkn strains. Two oral β-lactams were further evaluated in combination with first-line drugs for Mkn treatment, showing in vitro favorable interactions.

2.     Results

-          The title of table 1 should be written before the table not after the table.

-          The title should be rewritten provisionally and the authors should mention the four types of drugs, Penicillins, Cephalosporins, Carbapenems, Monobactam.

-          Page 3, 2.2 Amoxicillin/clavulanate and cefadroxil were the most active β-lactams against a panel of Mkn clinical strains; this should be written as a title not as sentence.

-          Despite that figures well presented, authors did not include any statistical analysis to compare the difference in the activities of different antibiotics. So, statistical analysis part should be included.

3.     Discussion

Well written, but need editing and some grammatical errors are there.

At the end of the discussion part, authors have to add the limitation of this study

Conclusion: could be improved.

4. Materials and Methods

4.4 Antibiotics Susceptibility Testing; in this subsection, authors should add the equations which illustrate how they calculate the MIC OR.

Same thing for 4.5 Time-Kill Assays

References:

Authors did not follow the journal guidelines during citation or formatting the references in the text or in the reference section. So, all references style must be corrected accordingly.

§  References must be numbered in order of appearance in the text (including citations in tables and legends) and listed individually at the end of the manuscript. We recommend preparing the references with a bibliography software package, such as EndNote, Reference Manager or Zotero to avoid typing mistakes and duplicated references. Include the digital object identifier (DOI) for all references where available.

§  Citations and references in the Supplementary Materials are permitted provided that they also appear in the reference list here.

§  In the text, reference numbers should be placed in square brackets [ ] and placed before the punctuation; for example [1], [1–3] or [1,3]. For embedded citations in the text with pagination, use both parentheses and brackets to indicate the reference number and page numbers; for example [5] (p. 10), or [6] (pp. 101–105).

For example

Author 1, A.B.; Author 2, C.D. Title of the article. Abbreviated Journal Name Year, Volume, page range

Author Response

Reviewer 2

Abstract: should be corrected

Point 2.1:  Authors should minimize the introduction and illustrates more about results and also, they have to rewrite the conclusion. 

Answer 1: The length of the introduction was reduced and the results and conclusion were modified.

Point 2.2:  Line 13-15 in the second sentence should be corrected (rewrite it). 

β-lactams are the most important class of antibiotics with an estimated of 65% of the global antibiotic market; although, they have been traditionally dismissed for the treatment of mycobacterial infections, as they were considered inactive against mycobacteria. 

Answer 2: The sentence was rewritten.

Although β-lactams are the most important class of antibiotics, representing the 65% of the global antibiotic market, they have been traditionally dismissed for the treatment of mycobacterial infections, as they were considered inactive against mycobacteria.” (Lines 14-16)

Point 2.3:  All the citation formats of the references in the text should be corrected according to the journal guidelines. 

Answer 3: References were corrected according to journal guidelines

Introduction: 

Point 2.4:  The first sentence lines 30-34 should be split into two sentences and rewritten. Also, the format style of references should be corrected according to journal style [1-4]. 

Mycobacterium kansasii (Mkn) is one of the most virulent and prevalent non-tuberculous mycobacteria (NTM), the most frequently found in immunocompetent patients and, in general, the second most prevalent cause of NTM disease in the United States, China, South American and some European countries [1]–[4]. 

Answer 4: Sentence was split and rewritten.

“Infection by Mycobacterium kansasii (Mkn) is the second most prevalent cause of non-tuberculous mycobacteria (NTM) disease in the United States, China, South Ameri-can and some European countries [1–4]. Mkn is one of the most virulent and prevalent NTM, being the most frequently found in immunocompetent patients  [5].” (Lines 33-36)

Point 2.5:  The second sentence lines 33-35, also needs correction and rephrasing. 

While diagnosis of most NTM- caused disease needs isolation of bacteria from at least two temporal independent sputum samples, because of the elevated pathogenicity of Mkn, a single positive culture may be enough evidence to initiate treatment. 

Answer 5: Sentence was rephrased.

In fact, because of its elevated pathogenicity a single positive culture may be enough evidence to initiate treatment, while diagnosis of most NTM-caused disease needs isolation of bacteria from at least two temporal independent sputum samples [3,6]”. (Lines 36-39)

Point 2.6:  Lines 44-46, should be corrected grammatically and the references should be corrected also. 

β-lactams are one of the largest groups of antibiotics available today with an excellent safety profile; they have been used to treat most infections caused by Gram-positive and 4 Gram-negative bacteria for the last 70 years [6], [7]. 

Answer 6: Sentence was corrected and rephrased.

“β-lactams are one of the largest groups of antibiotics available today with an excellent safety profile. Over the last 70 years, β-lactams have been used to treat most infections, either those caused by Gram-positive or by Gram-negative bacteria [10,11].” (Lines 52-54) 

Point 2.7:  The objectives of the study need to be rewritten and connecting words should be include for the second objective. 

In this study, we report the in vitro activity of several βlactams, including penicillins, cephalosporins, carbapenems and monobactams, against a panel of Mkn strains. Two oral β-lactams were further evaluated in combination with firstline drugs for Mkn treatment, showing in vitro favorable interactions. 

Answer 7: Sentence was rewritten and connected with the second objective

“In this study, we report the in vitro activity of several β-lactams, including penicillins, cephalosporins, carbapenems and monobactams, against a panel of Mkn strains. From this initial screening, two oral β-lactams were selected for further evaluation in combination with first-line drugs for Mkn treatment, showing in vitro favorable interactions.” (Lines 62-65)

Results 

Point 2.8:  The title of table 1 should be written before the table not after the table. 

- The title should be rewritten provisionally and the authors should mention the four types of drugs, Penicillins, Cephalosporins, Carbapenems, Monobactam. 

Answer 8: Table 1 title was situated before the table. Also, it was rewritten including penicillins, cephalosporins, carbapenems and a monobactam. It now reads:

Table 1. MIC (mg/L) of β-lactams against the Mkn ATCC 12478 strain. Penicillins, cephalosporins, carbapenems and a monobactam plus β-lactamase inhibitors were assayed against the Mkn ATCC 12478 reference strain in Middlebrook 7H9 broth plus ADC and glycerol and Cation Adjusted Muller-Hinton broth plus OADC. 1 β-lactamase inhibitors were tested at a fixed concentration of 4 mg/L.” (Lines 82-85)

Point 2.9:  - Page 3, 2.2 Amoxicillin/clavulanate and cefadroxil were the most active β-lactams against a panel of Mkn clinical strains; this should be written as a title not as sentence. 

Answer 9: Title was rewritten.

2.2 Amoxicillin/clavulanate and cefadroxil were validated as the most active β-lactams against a panel of Mkn clinical strains.” (Lines 88-89)

Point 2.10:  Despite that figures well presented, authors did not include any statistical analysis to compare the difference in the activities of different antibiotics. So, statistical analysis part should be included. 

Answer 10:

Figure 1 is a heat map representation of the MIC and data supporting the figure can be found in Table S1 where all the MIC data is provided. Figure 2 is a representation of the MIC values distribution and data supporting the figure are in Table S2. Figure 3, 4, 5 and S1 are time-kill kinetics assays, which are performed in duplicates and repeated at least twice. A sentence has been included in the figures to clarify this point. 

Discussion 

Point 2.11:  Well written, but need editing and some grammatical errors are there. At the end of the discussion part, authors have to add the limitation of this study 

Answer 11: We have corrected the grammatical errors and we have added the limitation of this study.

“This is the first study comprehensively reporting the activity of β-lactams alone and in combination against Mkn; however, it has some limitations. Our study is limited to extracellular, planktonic growth conditions. Other models reflecting the physiology of the bacteria at the site of infection such as slow or non-replicating, intracellular or biofilm growth conditions would contribute to obtain a more complete data set [57]. In addition, the concentrations used in the time kill assays were selected based on microbiological endpoints (MIC values) to evaluate the degree of in vitro interactions. Future studies will need to address the pharmacokinetic/pharmacodynamic (PK/PD) properties or serum concentrations reached at the site of infection of the compounds [24]. Novel modelling strategies coupled to dynamic in vitro PK/PD models such as the hollow fiber system might help to inform future clinical trials [58–60]”.  (Lines 360-370)

Conclusion: 

Point 2.12:  could be improved.

Answer 12: See above Point 1.10

Materials and Methods 

Point 2.13:  4.4 Antibiotics Susceptibility Testing; in this subsection, authors should add the equations which illustrate how they calculate the MIC OR. Same thing for 4.5 Time-Kill Assays 

Answer 13: The MIC is defined as the lowest drug concentration that inhibited conversion by 90%. We have added a paragraph to explain the % of MTT conversion is calculated. (Lines 430-440)

References: 

Point 2.14:  Authors did not follow the journal guidelines during citation or formatting the references in the text or in the reference section. So, all references style must be corrected accordingly. 

§ References must be numbered in order of appearance in the text (including citations in tables and legends) and listed individually at the end of the manuscript. We recommend preparing the references with a bibliography software package, such as EndNote, Reference Manager or Zotero to avoid typing mistakes and duplicated references. Include the digital object identifier (DOI) for all references where available. 

§ Citations and references in the Supplementary Materials are permitted provided that they also appear in the reference list here. 

§ In the text, reference numbers should be placed in square brackets [ ] and placed before the punctuation; for example [1], [1–3] or [1,3]. For embedded citations in the text with pagination, use both parentheses and brackets to indicate the reference number and page numbers; for example [5] (p. 10), or [6] (pp. 101–105). For example Author 1, A.B.; Author 2, C.D. Title of the article. Abbreviated Journal Name Year, Volume, page range

Answer 14: We have updated the references according to the journal guideline

Round 2

Reviewer 2 Report

Very thanking for your response. Still some minor changes need to be done.

-         Titles of tables and figures should be written according to the journal instructions.

Only name and number of figure should be bold, other no need.

Please follow this Example:- Table 1. MIC (mg/L) of β-lactams against the Mkn ATCC 12478 strain.

Figure 1. Heat map representation of the activity of thirteen selected β-lactams against Mkn 107 clinical strains.

-         Also, references should be corrected as the year should be bold. See references 8, 36, 53, 54. Please correct it

Author Response

Titles of tables and figures have now been changed according to the journal instructions.